# Pb Content, Risk Level and Primary-Source Apportionment in Wheat and Rice Grains in the Lihe River Watershed, Taihu Region, Eastern China

**DOI:** 10.3390/ijerph18126256

**Published:** 2021-06-09

**Authors:** Lian Chen, Shenglu Zhou, Qiong Yang, Qingrong Li, Dongxu Xing, Yang Xiao, Cuiming Tang

**Affiliations:** 1Sericultural & Agri-Food Research Institute, Guangdong Academy of Agricultural Sciences, Guangzhou 510610, China; dg1527004@smail.nju.edu.cn (L.C.); liqingrong@gdaas.cn (Q.L.); xingdongxu@gdaas.cn (D.X.); xiaoyang@gdaas.cn (Y.X.); tangcuiming@gdaas.cn (C.T.); 2School of Geography and Ocean Science, Nanjing University, 163 Xianlin Road, Nanjing 210023, China; 3Key Laboratory of Urban Agriculture in South China, Ministry of Agriculture and Rural Affairs, Guangzhou 510610, China

**Keywords:** grains, Pb contamination, primary-source apportionment, risk assessment, isotope analysis, model calculation

## Abstract

This study detailed a complete research from Lead (Pb) content level to ecological and health risk to direct- and primary-sources apportionment arising from wheat and rice grains, in the Lihe River Watershed of the Taihu region, East China. Ecological and health risk assessment were based on the pollution index and US Environmental Protection Agency (EPA) health risk assessment model. A three-stage quantitative analysis program based on Pb isotope analysis to determine the relative contributions of primary sources involving (1) direct-source apportionment in grains with a two-end-member model, (2) apportionment of soil and dustfall sources using the IsoSource model, and (3) the integration of results of (1) and (2) was notedly first proposed. The results indicated that mean contents of Pb in wheat and rice grains were 0.54 and 0.45 mg/kg and both the bio-concentration factors (BCF) were <<1; the ecological risk pollution indices were 1.35 for wheat grains and 1.11 for rice grains; hazard quotient (HQ) values for adult and child indicating health risks through ingestion of grains were all <1; Coal-fired industrial sources account for up to 60% of Pb in the grains. This study provides insights into the management of grain Pb pollution and a new method for its source apportionment.

## 1. Introduction

Pb is a heavy metal produced by the rapid economic development and greatly harmful to human body, especially to child. Human exposure to Pb occurs primarily through inhalation and ingestion, causing adverse health effects such as neurological and kidney diseases, even at low exposure levels [1,2,3,4], and Pb contamination in environmental compartments (rural soils, atmospheric dustfall, and food) has been partly documented in some articles [4,5,6,7,8,9]. Most investigations to date have focused on assessing contamination level and sources of Pb in soils and atmospheric particles [9,10,11,12], with few focusing on Pb contamination and its source apportionment in grains that are essential for metabolism. Rice and wheat are the world’s two most important cereal crops, contributing 45% of digestible energy and 30% of total protein in the human diet, as well as a substantial contribution to livestock feed [13]. More attention should therefore be given to these grains, particularly concerning risk assessment, the identification of contamination sources, and their relative contributions to grain Pb levels, with the aim of reducing the harm of such contamination through the development of pollution prevention and regulations.

A previous study indicated that the content of Pb in grains from 11 producing areas in China is higher than 1 mg/kg, in serious excess of the standard [14] and grain Pb contamination is the most serious among other heavy metals [15]. Zhang [16] evaluated the pollution level of wheat grains by utilizing the single factor pollution index in a mining area of Shanxi Province. It was found that the pollution index of Pb was about 3, far exceeding the red line (0.7). HQ through the ingestion of wheat grains was higher than 1 indicating eating wheat grains produced in this area has great health risk to human body. Ji et al. calculated the health risk of Pb contamination in soil, wheat, and vegetables around a mining area in Goseong, North Korea, and found that the health risk caused by rice ingestion was the highest, accounting for more than 75% [17].

However, compared with the above research on Pb contamination level and risk assessment of grains, relatively lower attention was focused on the Pb source apportionment in grains. Generally, Pb isotopic analysis is a more accurate source analysis method and has partly been applied in source apportionment for soils, dustfall, and grains [4,18,19,20]. However, due to the high cost of Pb isotopic analysis and the associated requirements for laboratory cleanliness, there have been relatively few studies of Pb source apportionment in soils and dustfall, and even fewer in grains. Pb has four stable isotopes: ^204^Pb, ^206^Pb, ^207^Pb, and ^208^Pb, and each Pb source has a distinct (sometimes overlapping) isotopic signature. Physicochemical and biological fractionation processes do not significantly alter these isotope ratios [18,21] so Pb isotopic compositions can ‘fingerprint’ Pb sources and trace Pb contamination in different environmental compartments [22]. Shang [23] determined Pb isotopic ratios in vegetable roots, stems, and leaves, and in soils and dustfall in Chengdu, China. The results indicated that ^206^Pb/^207^Pb ratios in vegetable leaves were very similar to those in dustfall, while ratios in roots were similar to those in soil. Previous studies have also found that Pb isotopic ratios in soils and vegetables in a mining area in Nanjing are significantly different, indicating that soil was not the main source of Pb in vegetables [24]. Zhao [25] applied Pb isotopic analysis to identify sources of Pb contamination in wheat grains in an industrial zone in Western Shanxi Province, and found that 95.5% of the Pb in wheat grains was derived from dustfall and 4.5% from soil. Soil and dustfall are thus the two main direct Pb sources for grains, with Pb entering the grain via root absorption from soil or uptake from dustfall via leaf stomata. To our knowledge, in the process of the source analysis of grain, most of the above studies only determined Pb the isotope ratios of grain and did not calculate the source contribution rate including direct source and primary source based on model analysis. This undoubtedly affects the understanding and control of Pb pollution in grain. According to the above analysis, soil and atmospheric dustfall are the two direct sources of grains. Further determination and analysis of Pb isotope in soils and dustfall is necessary to calculate the relative proportions of direct source of Pb in grains, and primary sources including anthropogenic and natural sources. Natural and anthropogenic primary sources of grains include coal-burning industries, vehicular emissions, sewage irrigation, fertilization, and so on, with primary sources for grains being almost identical to those of soil and dustfall. In this study, a three-stage quantitative analysis program based on Pb isotope analysis to determine the relative contributions of primary sources of grains was notedly first proposed. It contains three steps: direct-source apportionment in grains with a two-end-member model; apportionment of soil and dustfall sources using the IsoSource model; the integration of results of the first two steps through multiplication and addition. This three-stage quantitative model analysis can help us to understand the detailed process of grain Pb pollution, and also hereby guiding the further practices of prevention and control for grain Pb pollution.

The Lihe River watershed is located west of Taihu Lake, in the Southern Jiangsu Province. Taihu Lake is the third largest freshwater lake in China and lies in the lower reaches of the Yangtze River Delta, one of the most developed and populous regions of China. A previous study showed that median Pb concentration in soils of the lower Yangtze River Delta increased by 6.9% during 2004–2014 [26]. However, the Lihe River is an important water inlet of Taihu Lake. Its environmental pollution directly affects the water quality of the entirety of Taihu Lake. According to former studies, the potential ecological hazard index of heavy metal in soils in Lihe River estuary had reached a serious level (>220), making it the most contaminated among the estuaries of the 24 main rivers flowing into and out of Taihu Lake [27]. The main reason for this problem is that many types of industrial activity occur throughout Lihe River Watershed, including ceramics factories, refractory material plants, and chemical plants. High concentrations of Pb in soils in this region would definitely lead to increased concentrations in grains, but there has been little study on Pb contaminated level, risk assessment, and source apportionment in grains in this high-risk area.

In this study, Pb content level to risk assessment to sources apportionment was clearly and integrally described, in the Lihe River Watershed of the Taihu region, East China. Additionally, from the specific implementation details, a novel method was conducted for source apportionment of grains. The aims were to (1) determine Pb content levels and influencing factors in grains of Lihe River Watershed; (2) assess the ecological and health risks posed by Pb contamination in grains based on single factor pollution index and US EPA health risk assessment model; and (3) quantitatively analyze the direct- and primary-sources of grain Pb pollution based on three-stage quantitative analysis program. Through sample collection, experimental analysis, especially Pb isotope analysis, and model calculation, the results obtained from this study can provide insights into the management of grain Pb pollution and a new method for its source apportionment.

## 2. Methods

### 2.1. Field Sampling

Lihe River Watershed includes the towns of Hufu and Dingshu, with a surface area of ~260 km^2^ [28] (Appendix A). The total area of agricultural land is 57.8 km^2^, comprising dry (21.4 km^2^) and paddy (36.4 km^2^) land, where rice and wheat are cultivated. Contamination receptor samples (32 wheat grain and 32 rice grain), direct source samples of grain (32 soil and 10 atmospheric dustfall), and primary source samples of grain (32 irrigation water, 10 coal ash, 10 fuel-burning dust and 10 chemical fertilizers) were collected throughout the study area.

#### 2.1.1. Grains

Grain samples were collected from 32 randomly selected sites throughout the study area when the crops were ready for harvest; wheat in May and rice in October 2016 (Figure 1). Sixty-four crop samples, 32 each of wheat and rice ears, were collected with each sample comprising 5–9 subsamples with a total weight of 0.5–1.0 kg. After collection, the grains were cleaned with deionized water, oven-dried at 60 °C to constant weight, ground to <250 μm mesh in a pre-cleaned steel grinder and stored in polythene zip-lock bags [29].

#### 2.1.2. Soils

Soils in crop fields were sampled at 0–10 cm depth at each of the grain sampling sites, with each sample of 0.5–1.0 kg being divided into 5–9 subsamples. The samples were air-dried at room temperature, ground, and passed through a 2 mm nylon sieve to remove stones and plant roots [30]. The fine soil powders were stored in polythene zip-lock bags.

#### 2.1.3. Atmospheric Dustfall

Ten wet/dry dustfall monitoring points were set up in the form of a cross, based on the type of land use and urban layout: the NW–SW axis incorporated (sequentially) woodland, farmland, a suburban area, a town center, suburban area, and farmland; the NE–SW axis incorporated farmland, a town center, suburban area, and woodland. Monitoring quarterly was undertaken during 1 September 2016 to 1 September 2017 using a custom-made collection device (Appendix A); three replicate samples were collected at each sampling point. After collection, samples were evaporated to dryness, weighed, and analyzed for Pb concentrations and isotopic ratios.

#### 2.1.4. Primary Source Samples of Grain

Irrigation water samples (n = 32) were collected in fields next to the grain and soil sampling points, and at the same time. They were transferred to polyethylene bottles, nitric acid (0.1% *v*/*v*) was added as a preservative 23, and the bottles were stored at 20 °C pending analysis. Before analysis, 100 mL subsamples of each water sample were combined to give a 3.2 L total volume, which was filtered through pre-weighed 0.45 mm membrane filters to collect suspended particulate matter. Filters were oven dried to constant weight at 60 °C. Pb concentrations and isotopic ratios were determined for both suspended matter and filtered water.

Fuel-burning dust (i.e., particulate emissions from diesel- or gasoline-powered vehicles) was collected from vehicles in parking lots near the ten dustfall sampling sites using a clean toothbrush to extract residues from vehicle exhaust tailpipes. Coal-ash samples were also collected near the dustfall sampling sites using a toothbrush to collect dust from surfaces such as windowsills near each industrial facility. The dust samples were filtered through 200-mesh sieves, and subsamples of ~10 g were analyzed for Pb isotopes.

Fertilizer use by farmers was surveyed using a detailed questionnaire regarding types and amounts of fertilizer used annually. The results indicated that the study area could be divided into two sub-areas based on administrative regions. Large amounts of nitrogen, phosphate, potash, compound fertilizers, and organic fertilizers were applied in each sub-area. A sample of each type of fertilizer (10 in total) was obtained from selected retail stores for Pb isotopic analysis.

### 2.2. Sample Testing

The test indices were Pb total-concentration, acid-soluble Pb fractions, Pb isotopes and soil properties including particle size, pH, organic matter (OM), cation-exchange capacity (CEC), and electroconductivity (EC).

The acid-soluble Pb fractions of soil were analyzed by 0.1 mol L^−1^ HCl extraction. Pb total-concentration of soil and dustfall were digested using HCl-HNO_3_-HF-HClO_4_ and grains were digested using HNO_3_-HClO_4_-H_2_O_2_. They were then determined via inductively coupled plasma mass spectrometry (ICP-MS; Elan 9000; Perkin Elmer AB SCIEX, Redwood City, CA, USA). To ensure analytical accuracy, reference materials (GBW07405 for soil and dustfall, GBW07602 for grain), blanks, and duplicates were regularly analyzed (after every five sample measurements). The Pb isotope sample (^208^Pb/^206^Pb and ^206^Pb/^207^Pb) used for the sample pretreatment was the same as that used for Pb digestion. The Pb samples were diluted to ~30 µg L^−1^ before the ICP-MS analysis. Pb isotopic ratios were corrected using US National Institute of Standards and Technology (NIST) standard reference material SRM981.

Soil particle-size distribution was determined using a MasterSizer 2000 (Malvern Panalytical, Malvern, UK) laser particle-size analyzer [31]. Soil pH was determined by potential method; organic matter (OM) content was determined by loss on ignition; cation-exchange capacity (CEC) was determined by the ammonium acetate exchange method based on Forestry Industry Standards of the People’s Republic of China (LY/T1210–1275-1999); conductivity (EC) was determined using a Conductivity Analyzer [32].

### 2.3. Data Analysis

Statistical analyses, such as Pearson correlation analysis and significance testing of all data were carried out in SPSS 26.0 (IBM Corp.:Armonk, NY). In the significance testing, LSD multiple post-hoc comparisons of mean values by one-way analysis of variance (one-way ANOVA) were undertaken to determine the difference. In spatial analysis using ArcMap 10.0(Environmental Systems Research Institute (Esri), Redlands, California, USA) software, the inverse distance-weighting method was used for spatial interpolation. The results of the Pb isotope analysis were plotted in a coordinate system using Origin 8.5 (OriginLab, Northampton, MA, USA).

#### 2.3.1. Ecological Risk

The ecological risk of Pb pollution in soil, grain and dustfall was assessed here by the single factor pollution [33]. The calculation formula is as follows:*P_i_* = *C_s_/C_r_*(1)
where *P_i_* is the ecological risk of Pb in soil, grain and dustfall, *C_s_* is the concentration of Pb in the sample (soil, dustfall, or grain), and *C_r_* is their background values. For soil and dustfall samples, the *Cr* value is 26.2 mg/kg. For grains, *C_r_* takes the value of 0.4 mg/kg. Through calculation, when *P_i_* > 0.7, the larger the *P_i_*, the more serious the Pb pollution [34].

#### 2.3.2. Health Risk

The health risk assessment model developed by the EPA was used to assess the human-health risk of Pb pollution in grains through ingestion, based on average daily doses [35].

Health risk of Pb in grains is expressed as the hazard quotient (HQ) and it was calculated by the proportion of the amount of Pb ingested with contaminated grains to the reference oral dose (RfDo) for the local community.
(2)$$HQ=\frac{CDI}{RfDo}$$
(3)$$CDI=\frac{C\times IngR\times CF\times EF\times ED}{BW\times AT}$$
where CDI is the mass of the Pb contacted per unit bodyweight per unit time; RfDo is a safe estimate of the daily exposure of a human population The RfDo for Pb used in this study was 4.0 × 10^−3^ mg/kg [35]. Here, the parameter in Formula (3) was all shown in Appendix A. The health risk was divided into five grades according to the values of HQ [35], as shown in Appendix A.

#### 2.3.3. Source Apportionment Model

Source apportionment of grains to quantify relative contributions of pollution sources based on Pb isotopic data in this study were analyzed by two types of models: an end-member mixing model, and the EPA IsoSource model (Environmental Protection Agency, USA) [36]. The former was commonly used in earlier studies, with two or three end members [23,25,37], limiting pollution receptor modelling to two or three pollution sources, respectively. The greater the number of end members, the more calculation is required, and three-end-member calculations are lengthy and tedious with a limit of three possible pollution sources. IsoSource modelling effectively solves these problems [36].


(1)Two-end-member mixing model


For receptors with only two pollution sources, the contribution ratio of the sources can be calculated by the two-end-member mixing model with simple linear equations [25,38]. Here, this model was used for direct-source apportionment in grains using Equation (4), with there being only two direct Pb sources, soil and dustfall. If *f_a_* and *f_b_* are the Pb contributions of sources 1 and 2 to the contaminated samples; *R_g_*, *R_a_,* and *R_b_* are the values of ^207^Pb/^206^Pb (or ^208^Pb/^207^Pb) ratios of the contaminated wheat or rice grain, source 1, and source 2, respectively; the relative contributions of sources 1 and 2 can be calculated as:(4)$$\left\{\begin{array}{l} f_{a}=\frac{R_{g}-R_{b}}{R_{a}-R_{b}}\\ f_{b}=1-f_{a} \end{array}\right\}$$


(2)IsoSource model


The source contributions of soil or dustfall which have three or more sources, were calculated by IsoSource modelling. It was based on mass conservation principle of stable isotopes. The calculation formula is as follows [36,39]:(5)$$\left\{\begin{array}{l} I_{r}=\sum_{i=1}^{n}R_{i}\partial_{i}\\ R_{1}+R_{2}+R_{3}+\ldots \ldots +R_{n}=1 \end{array}\right\}$$
where *I_r_* is the isotopic ratio of contamination receptor (soil or dustfall), *R_i_* is the contribution rate of source *i*, and *∂_i_* is the isotopic ratio of source *i*. In this study, there were two isotopic systems (^206^Pb/^207^Pb and ^208^Pb/^206^Pb) and six (soil) or three (dustfall) potential sources, so the possible solutions of various combinations of the contribution rate range and mean value of the sources are obtained by using the above equations. Before running the model, two parameters need to be set: one is the source increment, which is generally set to 1%; the other is mass balance tolerance. If it is set to 0.1‰, it means that the difference between the sum of weighted isotopic values of each pollution source and the isotopic values of the acceptor does not exceed 0.1‰, then the proportional combination is considered as optimal solution [39].

## 3. Results and Discussion

### 3.1. Pb Levels and Risk in Soil and Dustfall

A wide range of Pb total concentrations was observed in soils across the study area (Table 1). The Pb concentrations exceeded background values in 94% of the 32 soil samples with mean Pb being higher than their respective background values. A comparison of the mean Pb concentrations with the Chinese standard for agricultural soil indicated that Pb did not exceed the standard value. The concentration of available Pb indicated their degree of mobility in soil. The mean ratio of available fraction to total concentration of Pb was 67.4%. Tao et al. used the same method to extract the available Pb in soil. The results showed that the mean ratio of available fraction to total concentration of Pb was about 20% [40]. Yang et al. studied the bioavailability and ecological risk assessment of heavy metals in soils around a mining area in Xinjiang, China and found that the ratio of available fraction to total concentration of Pb was 9% [41]. These results indicated that the mobility of Pb and therefore the greatest potential bioavailability in this study was relatively high.

Mean concentrations of Pb in dustfall was 223.86 mg/kg, significantly higher than background values. In comparison, concentration of Pb in Shijiazhuang City was 140 mg/kg [42], lower than that in this study. Also, Pb concentration in dustfall was similar with those measured in Shanghai [43]. The mean Pb concentration in the dustfall was significantly higher than background values, with almost nine times higher than the background value. These results all indicated relatively high degrees of contamination in dustfall of this region.

According to Equation (1) for ecological risk, the mean *P_i_* values of Pb for soils and dustfall were 1.52 and 8.54, respectively. The result indicated that Pb contamination level was higher in dustfall than in soil, with both exceeding the warning value (0.7), suggesting an extremely high ecological risk and pointing to the need for careful control of Pb pollution.

### 3.2. Pb Levels in Grains

The translocation of Pb from soils to grains can be described by the bio-concentration factor (BCF), reflecting the ability of plants to accumulate Pb. It is calculated as the ratio of Pb concentration in grain and soil [33]:

Wide ranges of Pb concentrations and BCF values were observed in grains across the study area (Figure 1). Multiple post-hoc comparisons of mean values by one-way analysis of variance were undertaken to determine the difference in Pb concentrations between the two grain types. Maximum, minimum, and mean Pb concentrations were higher in wheat grains than in rice grains, although Pb BCF values in the two grain types were similar. Concentrations of Pb in samples of both crops exceeded the Tolerance Limit for Pb in food (GB 2762-2005) set by the Ministry of Health of the People’s Republic of China (0.2 mg/kg) (Figure 1A) [29], indicating the seriousness of the pollution problem. Mean Pb BCF values in the two grain types were both <<1, indicating the activity of Pb in soil is very weak and its uptake by plants is generally low, also implying that only a very small part of Pb contamination in grains derived from soil.

In order to further identify the contamination-controlling factors of grains, response relationships between Pb levels in grains and environmental factors such as soil physicochemical properties and Pb concentrations in dustfall were investigated. This also can provide preliminary determination of pollution sources for grains. The detailed value of soil properties was presented in Appendix A.

According to the correlative analysis, there was no significant correlation between Pb concentration in wheat or rice grains and soil physicochemical properties (CEC, EC, OM, and particle size; there was some correlation with pH in the case of rice) (Table 2), consistent with results of a study of Pb contamination of wheat in Southern Jiangsu Province [34], nor was there any significant correlation between Pb concentration in these grains and total or available Pb concentrations in soil. However, there was a degree of correlation between Pb concentrations in grains and dustfall, implying Pb in wheat and rice may be derived mainly from dustfall rather than soil.

### 3.3. Pb Risk Assessment of Grain

The ecological risk pollution index of Pb in grains were calculated using Equation (1), with the values of 1.35 for wheat grains and 1.11 for rice grains, respectively, according to the limit standard of Pb in grain issued by China. This result indicated that the ecological risk of Pb pollution in wheat grains was relatively higher than that in in rice grains, and they both far exceeded the warning value (0.7), indicating an extremely high ecological risk and pointing to the need for careful control of Pb pollution.

Health risks associated with Pb exposure through ingestion of wheat and rice grain for children and adults were calculated using Equations (2) and (3), with results given in Table 3. However, contrary to ecological risk, the values of HQ in wheat grain were lower than that in rice grain. This may be related to the large amount of rice eaten every day. Although all HQ values were all <1, they were at 10^−1^ orders of magnitude, which indicates that the risks of Pb pollution in grains to human health are relatively low. However, the average concentrations (0.54 for wheat grain and 0.45 mg/kg for rice grain, respectively) were used for the calculation of HQ. If the maximum concentration of grains were used for calculation, the HQs of grains, especially for rice grains, could be close or even greater than 1. Thus, remediation should still be undertaken to prevent further risks. Also, worth noting is that HQs of Pb pollution for child were higher than that for adult. High blood Pb levels have an extremely negative effect on children’s intelligence.

### 3.4. Source Apportionment for Grain

#### 3.4.1. Direct-Source Apportionment for Grain

Our quantitative analyses indicate that soil and atmospheric dustfall are direct sources of Pb contamination in crop grains. Isotopic ratios of ^207^Pb/^206^Pb and ^208^Pb/^206^Pb in wheat and rice grains and their direct sources are plotted in Figure 2.

The mean Pb isotopic ratios of wheat and rice grains lie between those of dustfall and soil (Figure 2), implying that both may contribute to grain contamination, but with wheat and rice ratios being closer to dustfall ratios than to soil ratios, and therefore perhaps more greatly influenced by dustfall source. Compared with the ratios of rice grains, the wheat ratios are more similar to those of dustfall, which thus make the contribution of dustfall source was greater in wheat grains than in rice grains, whereas the contribution of soil source was just the opposite.

On further analysis, a linear relationship between Pb isotopic ratios in dustfall, soil, and wheat or rice grains is indicated in Figure 2, both satisfying conditions of the two-end-member model. For wheat grains, the values of ^206^Pb/^207^Pb for grain, dustfall, and soil were substituted as *R_g_*, *R_a_*, and *R_b_*, respectively, in Equation (1), with resulting fa and fb values of 93.6% and 6.4%, respectively, i.e., dustfall contributes 93.6% of wheat Pb contamination and soil 6.4%. Similarly, the application of ^208^Pb/^206^Pb ratios in Equation (1) yielded *f_a_* and *f_b_* values of 88.8% and 11.2%, respectively. The contribution of dustfall to Pb contamination of wheat was thus 88.8% and that of soil 11.2%, reasonably consistent with the ^206^Pb/^207^Pb results. The slight difference may be due to fluctuations in isotopic ratios; the two sets of results were averaged to yield dustfall and soil contributions to wheat Pb contamination of about 91% and 9%, respectively.

For the rice grains, the ^206^Pb/^207^Pb ratios in grains, dustfall, and soil were substituted into Equation (1) as *R_g_*, *R_a_*, and *R_b_*, respectively, yielding fa and fb values of 76.6% and 23.4%, respectively. Dustfall and soil thus make contributions of 76.6% and 23.4%, respectively. Similarly, ^208^Pb/^206^Pb ratios were applied in Equation (2), yielding dustfall and soil contributions of 78.8% and 21.2%, respectively, with the slight difference again likely being due to fluctuations in Pb isotopic composition. The two datasets were averaged, yielding dustfall and soil contributions to rice Pb contamination of about 78% and 22%, respectively.

Isotope analysis and model calculations here indicate that the contribution of dustfall is much greater than that of soil. Atmospheric dustfall was found to be the main source of Pb contamination in grain crops. Pb contamination of crops by dustfall should therefore be paid more attention. The same results have been verified in previous studies. Yang et al. [44] studied on the main sources of lead on grain crop (wheat) samples with isotopes and found that atmospheric fallout is a more significant source of Pb concentration in wheat grains than in soil. Dalenberg and Driel [45] have estimated that dustfall contributes > 90% of Pb contamination to wheat grains. Shang [23] studied Pb isotopic compositions of above-ground (stem and leaf) and below-ground (root) parts of vegetables including radish, lettuce, ginger and chives in the Chengdu area, with results indicating that ^206^Pb/^207^Pb ratios of vegetable leaves were very similar to those of dustfall, with the Pb isotopic composition of rhizosphere soil also being similar to that of vegetable roots. Pb in plant roots originates mainly from rhizosphere soil, with less migrating to leaves and grains. Plant leaves, on the other hand, generally absorb Pb from polluted air through leaf stomata, with leaves and above-ground tissues having Pb isotopic compositions more similar to that of dustfall. Hu et al. [24], in a study of Pb isotopic compositions of vegetables and soils near a mining area in Nanjing, found large differences between the two, again indicating that soil is not the main source of Pb in vegetable leaves. Zhao [25], in a study of an industrial zone in Shaanxi Province, determined contributions of pollution sources by isotopic analysis and again found that dustfall is the main source of Pb contamination in wheat grains. Past research has showed that the activity of Pb in soil is very low [46] and its uptake by plants is generally low [47], with the amount transferred from roots to grains being very small [46,48]. Feng et al. [10] investigated metals (Pb, Cd, Cr, Zn, and Cu) concentrations and stable Pb isotope ratios in rice plants exposed and unexposed to highway traffic pollution in Eastern China and found that Pb in rice plants were partly from the foliar uptake and considerable amount of foliar absorbed Pb and Cd were transported to rice grain. This partially explains the reasons for such results. However, it should be noted that most previous studies did not calculate the contribution rates of dustfall and soil to grain Pb, respectively. In this study, the calculation and display of the results are realized. In addition, through calculation, the contribution of dustfall to wheat (~91%) was higher than that to rice (~78%), possibly due to differences in plant absorption mechanisms, or to the husk structure of the two crops; the wheat structure may allow greater penetration of dustfall.

#### 3.4.2. Primary-Source Apportionment for Grain

Dustfall and soil are the two direct sources of grain Pb. In order to obtain the primary source of grains, the Pb pollution sources of soil and dustfall need to be further apportioned.

##### Source Apportionment for Soil and Dustfall

Consistent with source apportionment of grain Pb pollution, the first step is to screen out sources not related to soil contamination by using the (^208^Pb/^206^Pb)–(^207^Pb/^206^Pb) plot for contaminated soils and all groups of possible Pb sources in origin 8.5, and then the IsoSource model was applied to determine source contributions (Figure 3). We applied the Pb isotopic compositions of Ningwu belt granite (in the Yangtze River area), with ^207^Pb/^206^Pb and ^208^Pb/^206^Pb ratios of 0.844 and 2.081, respectively [49], to represent the geological source Pb ratios. The isotope ratios for pollution source of soil and pollution receptor were all displayed in Figure 3. It was generally shown that Pb isotopes ratios of the same substance have no significant difference and are clustered together, such as soil and dustfall. However, Pb ratios of chemical fertilizer of various sorts have a scattered distribution, consistent with their origin from diverse sources. The Pb isotopic ratios of the 32 soil are different from those of irrigation water and phosphate fertilizer in the red dotted box, indicating that these sources did not contribute significantly to soil Pb contamination. In contrast, dustfall samples, geological sources, water suspended matter, and nitrogen, potash, and organic fertilizers are clustered near the soil cluster in the black solid box (Figure 3), implying that these were more likely significant contributors to soil contamination. Nevertheless, it is noticed that compound fertilizer Pb ratios (^207^Pb/^206^Pb = 0.769 ± 0.004; ^208^Pb/^206^Pb = 1.863 ± 0.002) were very different to those of our samples and are not included in Figure 3; such fertilizers may thus not contribute Pb to soils.

The pollution sources including geological source, dustfall samples, suspended matter in water, nitrogen fertilizers, potash fertilizers, and organic fertilizers are substituted into the IsoSource model after preliminary screening. After operation calculation, the result was obtained that the dustfall mean contribution to soil Pb contamination was about 66.7%, with dustfall thus being the major contributor to soil Pb contamination. N, K, and O fertilizers together contributed 18%, while geological sources and irrigation water each contributed a marginal ~10% (Figure 3).

Similarly, Pb isotopic ratios of potential contamination sources for dustfall including coal dust, vehicular emission, and geological sources are displayed in Appendix A with origin 8.5 software. Geological source ratios were 0.844 and 2.081 for ^207^Pb/^206^Pb and ^208^Pb/^206^Pb, respectively; and coal-dust ratios were 0.870 and 2.130, respectively. Chinese coal has a wide range of Pb isotopic ratios (^207^Pb/^206^Pb = 0.850–0.925), with values being higher in southern China than in the north [18,50,51]. Vehicular emission ^207^Pb/^206^Pb and ^208^Pb/^206^Pb ratios averaged 0.885 and 2.069, respectively, consistent with values recorded in an industrial area (0.87 and 2.02, respectively) [36]; and in the Pearl River Delta (0.862 and 2.085, respectively [52]. Pb source contributions based on IsoSource are given in Figure 3. The mean contribution order of dustfall Pb was ranked as coal dust (69%) > geological sources (23%) > vehicular emissions (8%), which thus make coal dust the major source and vehicular emissions a marginal source.

##### Primary-Source Apportionment Progress for Grains

The contributions of direct sources (i.e., dustfall and soil) to wheat or rice Pb contamination were estimated above using the two-end-member mixing model, and this was further applied in calculating primary-source apportionments with the process and results as shown in Figure 4 and Figure 5.

The contribution of direct dustfall to wheat Pb contamination is about 91% (Section 3.4.1), while source apportionment to dustfall itself includes coal-fired industries, 69.2%; vehicular emissions, 8.2%; and geological sources, 22.6% (Section “Source apportionment for soil and dustfall”). The multiplication of these datasets indicates that coal-fired industries, vehicular emissions, and geological sources contribute, via dustfall, 63.1%, 7.5%, and 20.6%, respectively, of wheat Pb contamination. Soil makes a direct contribution of about 9% to wheat Pb contamination (Section 3.4.1), with contributions to soil itself being: dustfall, 66.7%; geological sources, 8%; irrigation water, 7.3%; nitrogen fertilizer, 6.1%; potassium fertilizer, 6.5%, and organic fertilizer, 5.5% (Section “Source apportionment for soil and dustfall”). By multiplication again, the contributions of dustfall, geological sources, irrigation water, and nitrogen, potassium, and organic fertilizers to Pb contamination of wheat, via soil, are 5.9%, 0.7%, 0.6%, 0.5%, 0.6%, and 0.5%, respectively. The dustfall source was further unraveled, by the same method, to yield contributions from coal-fired industries, vehicular emissions, and geological sources, via dustfall, of 4.1%, 0.5%, and 1.3%, respectively. Finally, all types of primary source were combined to give their relative contributions to wheat Pb contamination, ranked as coal-fired industries (67.2%) > geological sources (22.6%) > vehicular emissions (8%) > potassium fertilizer (0.6%) ≈ irrigation water (0.6%) > organic fertilizer (0.5%) ≈ nitrogen fertilizer (0.5%).

For rice grains, the contribution of the direct source dustfall was about 78% (Section 3.4.1). The source apportionment of Pb contamination in dustfall was ranked as coal-fired industries, 69.2%; vehicular emissions, 8.2%; and geological sources, 22.6% (Section “Source apportionment for soil and dustfall”). similarly, multiplication of the two datasets indicates contributions, via dustfall, of 53.8%, 6.4% and 17.6% for coal-fired industries, vehicular exhausts, and geological sources, respectively. The direct-source soil to rice contamination was about 22% (Section 3.4.1), including contributions of dustfall (66.7%), geological (8%), irrigation water (7.3%), and nitrogen (6.1%), potassium (6.5%), and organic (5.5%) fertilizers (Section “Source apportionment for soil and dustfall”). Using the same method, the contribution rates of these sources to rice contamination were 14.9%, 1.8%, 1.6%, 1.4%, 1.4%, and 1.2%, respectively. The dustfall source was further unraveled, yielding contributions from coal-fired industries, vehicular emissions, and geological sources of 10.3%, 1.2%, and 3.4%, respectively. Finally, all types of primary source were combined to yield relative rice Pb contamination source contributions of coal-fired industries (64.1%) > geological sources (22.8%) > vehicular emissions (7.6%) > irrigation water (1.6%) > potassium fertilizer (1.4%) ≈ nitrogen fertilizer (1.4%) > organic fertilizer (1.2%).

The above analyses of primary contamination sources for both wheat and rice grains indicate that coal-fired industries account for the largest proportion, of up to 67%, followed by vehicular emissions at about 8%. Chemical fertilizers and irrigation water contribute only up to 2%. It follows that control of coal-fired industries is the most important means of reducing Pb contamination of grain crops.

#### 3.4.3. Summary about Three-Stage Quantitative Analysis Results

In this study, we provided a new attempt to determine the relative contributions of primary sources in grains based on Pb isotope analysis and model calculation using a three-stage quantitative analysis program involving (1) direct-source apportionment in grains with a two-end-member model, which can determine the contribution rate of soil and dustfall Pb contamination to grain; (2) source apportionment for soil and dustfall using the IsoSource (Environmental Protection Agency, USA) model; and (3) integration of results of (1) and (2) through multiplying and adding to finally get the contribution rate of all primary-sources for grains.

In the first process of direct source apportionment of grains, two-end-member model was commonly used for pollution receptors with only two pollution sources. Pb in pollution receptors can be regarded as a mixture of two main sources and the relative contribution rate of pollution sources to Pb in pollution receptors can be calculated according to two-end-member model. In this study, direct-source apportionment in wheat or rice grains, which there were only two direct Pb sources, soil and dustfall, was calculated by the two-endmember. For wheat grains, the source contribution rates of dustfall and soil to grains Pb pollution were respectively ~91% and ~9% and ~78% and ~22% for rice grains. Atmospheric dustfall was found to be the main direct source of Pb contamination in grain crops.

In the second step of source apportionment for soil and dustfall, which respectively have six and three sources, IsoSource model was used for the calculation of source contribution rate. For soil, the mean contribution rate of atmospheric dustfall was highest (66.7%), with dustfall thus being the major contributor to soil Pb contamination. Geological sources and irrigation water each contributed a marginal ~8%. K (6.5%), N (6%), and O fertilizers (5.5%) together contributed 18%. For dustfall, the mean order of source contribution rates was ranked as coal dust (69%) > geological sources (23%) > vehicular emissions (8%), which thus make coal dust the major source and vehicular emissions a marginal source.

In the third process of primary-source apportionment for grains, sequential multiplication and addition were used for this calculation. For wheat grains, all types of primary source to wheat Pb contamination, ranked as coal-fired industries (67.2%) > geological sources (22.6%) > vehicular emissions (8%) > potassium fertilizer (0.6%) ≈ irrigation water (0.6%) > organic fertilizer (0.5%) ≈ nitrogen fertilizer (0.5%). For rice grains, all types of primary source to rice Pb contamination source contributions of coal-fired industries (64.1%) > geological sources (22.8%) > vehicular emissions (7.6%) > irrigation water (1.6%) > potassium fertilizer (1.4%) ≈ nitrogen ferti-lizer (1.4%) > organic fertilizer (1.2%). These results indicated that source of coal-fired industries account for the largest proportion, of up to 60%. The control of coal-fired industries is the most important means of reducing the Pb contamination of grain crops.

Through three-stage quantitative model analysis, not only the contribution rates of direct sources for grain were obtained, but also the contribution rates of primary sources for grain were calculated clearly. The results of each step obtained in the three steps can make us understand the detailed process of grain Pb pollution, and hereby guiding the further practices of prevention and control for grain Pb pollution.

## 4. Conclusions

Pb content level, ecological and health risk assessment, and direct and primary sources of wheat and rice grains in Lihe River Watershed were systematically and creatively investigated in this study. A three-stage quantitative analysis program, it should be noted, was first proposed to determine contributions of primary sources to Pb contamination of wheat and rice grains. The results obtained are as follows:

Average concentrations of Pb in wheat and rice grains were 0.54 and 0.45 mg/kg respectively, exceeding the Tolerance Limit for Pb in food (0.2 mg/kg). Pb content in grains showed no significant correlation with soil properties, conversely, a degree of correlation with dustfall, implying Pb in grains may be derived mainly from dustfall rather than soil. The BCF values of the two grains were both far less than 1, also suggesting the activity of Pb in soil is very low and its uptake by plants is generally low.

The ecological risk pollution indices of Pb were 1.35 for wheat grains and 1.11 for rice grains respectively, both far exceeding the warning value (0.7). HQ value indicating health risks through ingestion of wheat and rice grain for children and adults were at 10^−1^ orders of magnitude, <1. Risk assessment results indicated that the risks of Pb pollution in grains to human health are relatively low, although ecological risk of grain Pb pollution is high, and remediation should still be undertaken to prevent further risks.

Direct-source apportionments were estimated using a two-end-member model, with dustfall and soil contributing respectively ~91% and ~9% for wheat grains and ~78% and ~22% for rice grains. Primary-source apportionment for wheat and rice were obtained by integrating results of the first two steps. Primary Pb sources for wheat grains were ranked as coal-fired industries (67.2%) > geological sources (22.6%) > vehicular emissions (8%) > potassium fertilizer (0.6%) ≈ irrigation water (0.6%) > organic fertilizer (0.5%) ≈ nitrogen fertilizer (0.5%); while for rice contamination the rankings were 64.1%, 22.8%, 7.6%, 1.6%, 1.4%, and 1.2%, respectively. Coal-fired industries account for the largest Pb contamination contribution for the two grains of up to 67%. The control of coal-fired industrial sources may be the most important means of controlling grain Pb contamination.

## Figures and Tables

**Figure 1 ijerph-18-06256-f001:**
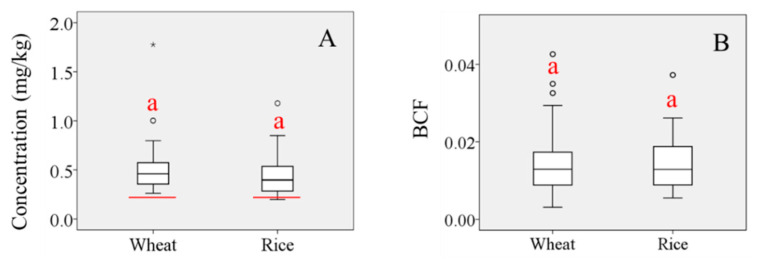
Box plots of (**A**) Pb concentration in wheat and rice grains (red line represents the Tolerance Limit for Pb in food, GB 2762-2005; “°” represents the discrete value; “*” represents extreme value); (**B**) BCF values for wheat and rice grains. (‘a’ indicates no significant difference at the *p* = 0.05 level).

**Figure 2 ijerph-18-06256-f002:**
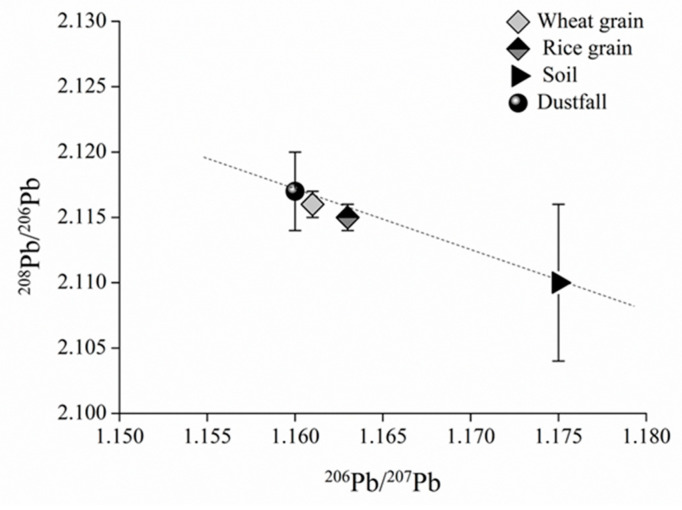
(^206^Pb/^207^Pb)–(^208^Pb/^206^Pb) diagram for dustfall, soil, and wheat or rice grains.

**Figure 3 ijerph-18-06256-f003:**
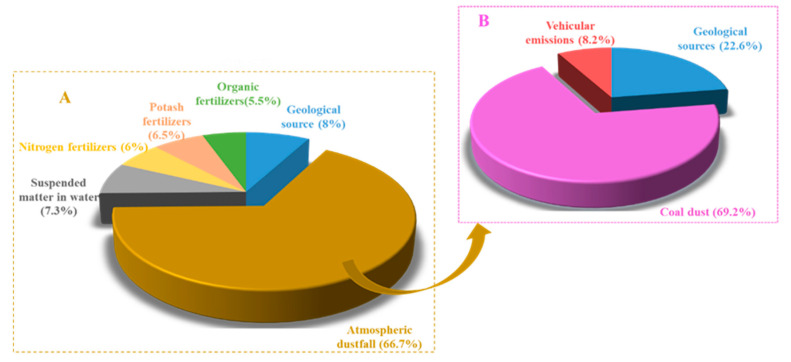
Relative Pb source contributions to soil (**A**) and dustfall (**B**) respectively based on IsoSource.

**Figure 4 ijerph-18-06256-f004:**
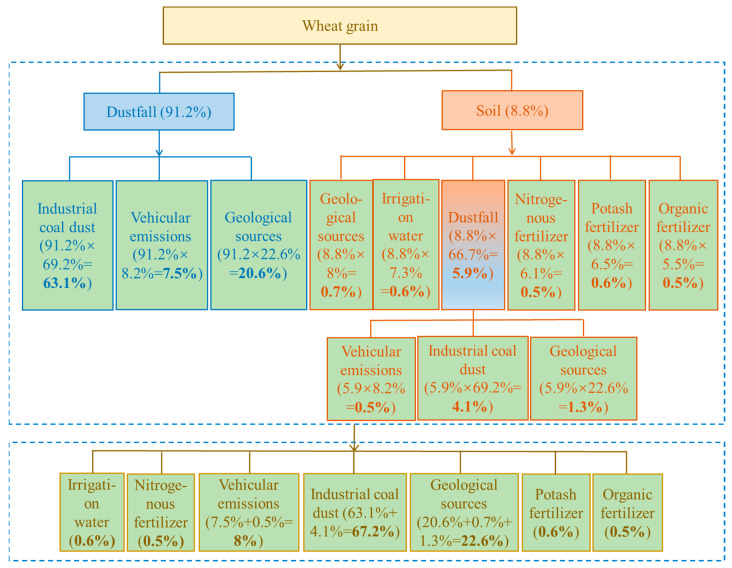
Analytical process and calculation result for primary source apportionment of Pb contamination of wheat grains.

**Figure 5 ijerph-18-06256-f005:**
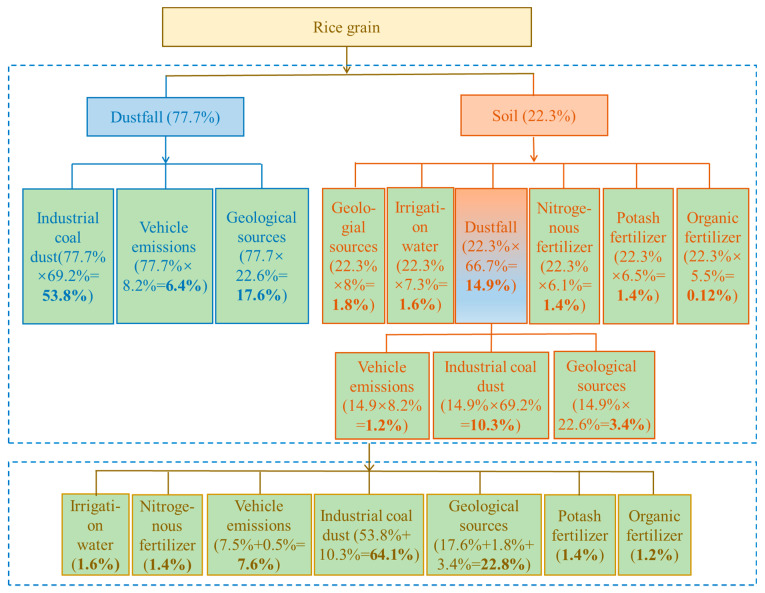
Analytical process and calculation results for primary source apportionment for Pb contamination of rice grains.

**Table 1 ijerph-18-06256-t001:** Pb concentrations and *P_i_* in soil and dustfall in the study area (mg/kg).

Parameter	Samples		Minimum Value	Maximum Value	Mean	SD
Concentration	Soil (n = 32)	Pb-s	21.26	141.72	39.78	20.61
		Pb-sa	12.51	91.84	26.84	13.61
	Dustfall (n = 10)		73.82	328.97	223.86	76.33
	Background values of Jiangsu		-	-	26.2	-
	Chinese standard for agriculture soil		-	-	300	-
*P_i_*	Soil (n = 32)		0.81	5.41	1.52	0.79
	Dustfall (n = 10)		2.82	12.56	8.54	2.91

Note: Pb-s = concentration in soil; Pb-sa = available Pb in soil; Pb-d = concentration in dustfall; “-” represents no reference value.

**Table 2 ijerph-18-06256-t002:** Correlation coefficients for Pb concentrations in wheat and rice grains versus soil properties and concentrations in soil and dustfall.

Grain Type	Pb-s	Pb-sa	Pb-d	pH	CEC	EC	OM	Mean Particle Size
Wheat	0.045	0.063	0.426 *	–0.135	–0.014	0.039	–0.346	0.027
Rice	0.036	0.214	0.360 *	0.407 *	–0.023	0.126	0.187	–0.076

Note: Pb-g = Pb concentration in grain; Pb-s = concentration in soil; Pb-sa = available Pb in soil; Pb-d = concentration in dustfall; “*” indicates significant correlation at the *p* = 0.05 level.

**Table 3 ijerph-18-06256-t003:** Mean values of hazard quotient (HQ) of Pb pollution through ingestion of wheat or rice grain.

HQ	Child	Adult
Wheat grain	2.67 × 10^−1^	2.43 × 10^−1^
Rice grain	5.12 × 10^−1^	4.70 × 10^−1^

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
