# Peer review of "Pb Content, Risk Level and Primary-Source Apportionment in Wheat and Rice Grains in the Lihe River Watershed, Taihu Region, Eastern China"

_ijerph, 2021, doi:10.3390/ijerph18126256_

Round 1
Reviewer 1 Report
Dear Authors,
First of all, I congratulate the authors for the interesting work entitled "Pb content, risk level and primary-source apportionment in wheat and rice grains in the Lihe River watershed, Taihu region, eastern China".
The article is interesting and well-written. It's logically arranged and carefully drafted. The results of the research have been correctly described. Formulated conclusions are correct.
I consider that the work is relevant in practice, and that it is well presented. Now I have some minor suggestions to improve the manuscript:
- Line 143. - 2.2. Data analysis - provide more information on the statistical methods used, e.g. level p, what kind of tests - Multiple post-hoc
- Line 213: is g kg–1 , Figure 2 is g/kg - it should be the same in the text
- Line 305 - Zhao [25] , Line 295 - Dalenberg and Driel –[39], Line 303 – Hu et. [24] - not in the References section. Check other literature
- Figure 2: - Correct the inscriptions, Y axis
Author Response
General comments:
First of all, I congratulate the authors for the interesting work entitled "Pb content, risk level and primary-source apportionment in wheat and rice grains in the Lihe River watershed, Taihu region, eastern China".
The article is interesting and well-written. It's logically arranged and carefully drafted. The results of the research have been correctly described. Formulated conclusions are correct.
I consider that the work is relevant in practice, and that it is well presented. Now I have some minor suggestions to improve the manuscript:
Response: We really appreciate that you give us positive comments and number of valuable suggestions. We have responded to the four comments and questions in a point-wise manner. Please check it in the manuscript for more details. The amended contents have been marked in red.
Point 1: Line 143. - 2.2. Data analysis - provide more information on the statistical methods used, e.g. level p, what kind of tests - Multiple post-hoc.
Response 2: Thank you very much for your valuable suggestion. LSD multiple post-hoc comparisons of mean values by one-way analysis of variance (one-way ANOVA) were undertaken to determine the level of p. We have added this information in the section of “Data analysis” in red text.
Point 2: Line 213: is g kg–1 , Figure 2 is g/kg - it should be the same in the text.
Response 2: Thank you very much for your significant question. According to your valuable suggestion, we have changed the mg kg–1 in line 163 and 213 to mg/kg to ensure the same in the full text. The revised text is marked in red.
Point 3: Line 305 - Zhao [25] , Line 295 - Dalenberg and Driel –[39], Line 303 – Hu et. [24] - not in the References section. Check other literature
Response 3: Thank you very much for your extremely valuable suggestion. We have checked all the references in the full text and modified them.
Point 4: Figure 2: - Correct the inscriptions, Y axis.
Response 4: Thank you very much for your significant questions. We have corrected the inscriptions, Y axis in figure 2 and redrawn it. The new version was shown in the text.

Reviewer 2 Report
Comments to the Author:
The authors of this paper present an interesting study on Pb content, risk level and primary-source apportionment in wheat and rice grains in the Lihe River watershed, Taihu region, eastern China. Nevertheless, this work might be polished up, taking into account the two following suggestions:
-Lines 117 : Please explain further how and why you made the choice of 32 samples from rice and 32 from wheat.
-Lines 169-171repeated in lines 178-180.
To conclude, reported data sufficiently presented, and the results supported the author’s conclusion adequately. Therefore, I think that this paper is suitable for publication after the correction of the above observation.
Author Response
General points:
The authors of this paper present an interesting study on Pb content, risk level and primary-source apportionment in wheat and rice grains in the Lihe River watershed, Taihu region, eastern China. Nevertheless, this work might be polished up, taking into account the two following suggestions:
To conclude, reported data sufficiently presented, and the results supported the author’s conclusion adequately. Therefore, I think that this paper is suitable for publication after the correction of the above observation.
Response: We really appreciate that you give us positive comments and number of valuable suggestions. We have responded to all the comments and questions in a point-wise manner. Please check it in the manuscript for more details. The amended contents have been marked in red.
Point 1: Lines 117 : Please explain further how and why you made the choice of 32 samples from rice and 32 from wheat.
Response 1: Thank you very much for your valuable problem. In this study, the method of selecting sampling points is as follows: Firstly, using the function of “create fishnet”in ArcGIS software to generate regular grid. According to the size of the study area, we set the height and width of the grid pixel as 2000 * 2000, and a total of 70 points were obtained when taking a point on each grid (as shown in figure 1 below). In other words, under ideal conditions, 70 samples were selected in this study. However, when we took the GPS coordinates of these 70 points to the study area for sampling, we found some of the 70 sampling points were not agricultural soil samples, or even if they were agricultural sites, the plant cultivated were vegetables or garden trees instead of rice or wheat. Thus, it is necessary to make appropriate adjustment around the preset sampling points according to the actual environment. And finally, the remaining 32 points can be collected. The actual coordinate position of the sampling points was recorded by handheld GPS positioning instrument, and the distribution map of actual sampling point position was generated by ArcGIS (figure 2).
Figure 1. The preset sampling point of the study area
Figure 2. The actual sampling point of the study area
Point 2: Lines 169-171 repeated in lines 178-180.
Response 2: Thank you very much for your significant question. We have deleted the repeated sentences in line 178-180 in the text.

Reviewer 3 Report
General comments
1. The experimental design is complete. It is damaging that only the soil has been analysed. Grain results should be validated by analysis
2. Similar publications in their theme and in their development exist and are quite old
Yang, J., Chen, T., Lei, M. et al. (2015). https://doi-org.ressources-electroniques.univ-lille.fr/10.1007/s11356-015-4601-9
Feng J F, et al. 2011. Source attributions of heavy metals in rice plant along highway in Eastern China. Journal of Environmental Sciences, 23(7): 1158–1164
3. The article is well written, easy to read and logically organized
4. The discussion around the contribution induced by the 3-stage quantitative model could be deepened, for example by a comparison of the results obtained with the 2 types of models. This would have made it possible to highlight the precision obtained
Detailed comments
Introduction
The bibliography must be deepened. It will also support the discussion
Lines 40: there are a lot of publications on industrial sites too
Materials and methods
All the analysis protocols for non-soil samples are missing
How atmospheric dustfall have been collected?
For statistics, the technical aspect is described. On the other hand, the description and the justification of the statistical tests carried out are missing.
Line 153: what is the reference?
Results and discussion
Lines 166 and following and lines 175 and following: are the same paragraphs with different references
Line 206: How were the concentrations in the grains analysed?
Line 208: it would be useful to know Pb-s, Pb-Sa, Pb-d
It would be beneficial for you to indicate the results of the concentrations in the soils and to comment on them in terms of the degree of contamination.
Figure 1: How do the 2 components of the ratio vary with respect to each other?
Line 225: How the concentrations of Pb in dust were analysed
Line 253: Indeed, lead is a cumulative toxicant
Line 257: For the calculation it looks like you have chosen the average concentration (0.5 and 0.4 mg / kg, respectively). It seems that for the maximum concentration, the HQ for rice could be close or even greater than 1
Lines 298: which vegetables? This influences a lot, depending on the very different accumulation capacities
Line 343: it is necessary to describe what was done
Author Response
Response to Reviewer 3 Comments
General comments:
Point 1. The experimental design is complete. It is damaging that only the soil has been analysed. Grain results should be validated by analysis
Response 1: Thank you very much for your significant question. We have added the analysis method of grain and dustfall in the section of “2.1 field sampling”. The added contents have been marked in red.
2.1. Field Sampling
Lihe River Watershed includes the towns of Hufu and Dingshu, with a surface area of ~260 km2 (Li, Huang et al. 2006) (Fig. S1). The total area of agricultural land is 57.8 km2, comprising dry (21.4 km2) and paddy (36.4 km2) land, where rice and wheat are cultivated. Contamination receptor samples (32 wheat grain and 32 rice grain), direct source samples of grain (32 soil and 10 atmospheric dustfall), and primary source samples of grain (32 irrigation water, 10 coal ash, 10 fuel-burning dust and 10 chemical fertilizers) were collected throughout the study area.
2.1.1 Grains
Grain samples were collected from 32 randomly selected sites throughout the study area when the crops were ready for harvest; wheat in May and rice in October, 2016 (Fig. 1). Sixty-four crop samples, 32 each of wheat and rice ears, were collected with each sample comprising 5–9 subsamples with a total weight of 0.5–1.0 kg. After collection, the grains were cleaned with deionized water, oven-dried at 60°C to constant weight, ground to <250 μm mesh in a pre-cleaned steel grinder, and stored in polythene zip-lock bags (Chen, Zhou et al. 2017).
2.1.2 Soils
Soils in crop fields were sampled at 0–10 cm depth at each of the grain sampling sites, with each sample of 0.5–1.0 kg being divided into 5–9 subsamples. Samples were air-dried at room temperature, ground, and passed through a 2 mm nylon sieve to re-move stones and plant roots (Lin, Wu et al. 2016). The fine soil powders were stored in polythene zip-lock bags.
2.1.3 Atmospheric dustfall
Ten wet/dry dustfall monitoring points were set up in the form of a cross, based on the type of land use and urban layout: the NW–SW axis incorporated (sequentially) woodland, farmland, a suburban area, a town center, suburban area, and farmland; the NE–SW axis incorporated farmland, a town center, suburban area, and woodland. Monitoring quarterly was undertaken during 1 September 2016 to 1 September 2017 using a custom-made collection device (Fig. S1); three replicate samples were collected at each sampling point. After collection, samples were evaporated to dryness, weighed, and analyzed for Pb concentrations and isotopic ratios.
2.1.4 primary source samples of grain
Irrigation water samples (n = 32) were collected in fields next to the grain and soil sampling points, and at the same time. They were transferred to polyethylene bottles, nitric acid (0.1% v/v) was added as a preservative 23, and the bottles were stored at 20℃ pending analysis. Before analysis, 100 mL subsamples of each water sample were combined to give a 3.2 L total volume, which was filtered through pre-weighed 0.45 mm membrane filters to collect suspended particulate matter. Filters were oven dried to constant weight at 60℃. Pb concentrations and isotopic ratios were determined for both suspended matter and filtered water.
Fuel-burning dust (i.e., particulate emissions from diesel- or gasoline-powered vehicles) was collected from vehicles in parking lots near the ten dustfall sampling sites using a clean toothbrush to extract residues from vehicle exhaust tail-pipes. Coal-ash samples were also collected near the dustfall sampling sites using a toothbrush to collect dust from surfaces such as window-sills near each industrial facility. Dust samples were filtered through 200-mesh sieves, and subsamples of ~10 g were analyzed for Pb isotopes.
Fertilizer use by farmers was surveyed using a detailed questionnaire regarding types and amounts of fertilizer used annually. Results indicated that the study area could be divided into two sub-areas based on administrative regions. Large amounts of nitrogen, phosphate, potash, compound fertilizers, and organic fertilizers were applied in each sub-area. A sample of each type of fertilizer (10 in total) was obtained from selected retail stores for Pb isotopic analysis.
2.2. Sample testing
The test indices were Pb total-concentration, acid-soluble Pb fractions, Pb isotopes and soil properties including particle size, pH, organic matter (OM), cation-exchange capacity (CEC), electroconductivity (EC).
The acid-soluble Pb fractions of soil were analyzed by 0.1 mol L–1 HCl extraction. Pb total-concentration of soil and dustfall were digested using HCl-HNO3-HF-HClO4 and grains were digested using HNO3-HClO4-H2O2. And then were determined via inductively coupled plasma mass spectrometry (ICP-MS; Elan 9000; Perkin Elmer AB SCIEX, Redwood City, CA, USA). To ensure analytical accuracy, reference materials (GBW07405 for soil and dustfall, GBW07602 for grain), blanks, and duplicates were regularly analysed (after every five sample measurements).
Point 2. Similar publications in their theme and in their development exist and are quite old
Yang, J., Chen, T., Lei, M. et al. (2015). https://doi-org.ressources-electroniques.univ-lille.fr/10.1007/s11356-015-4601-9
Feng J F, et al. 2011. Source attributions of heavy metals in rice plant along highway in Eastern China. Journal of Environmental Sciences, 23(7): 1158–1164
Response 2: Thank you very much for your valuable reminder. We have carefully read the above two references, which are indeed very relevant to our research, and then cited them in our paper.
Yang et al. (2015) studied on the main sources of lead on grain crop (wheat) samples with isotopes and found that atmospheric fallout is a more significant source of Pb concentration in wheat grains than in soil.
Feng et al. (2011) investigated metals (Pb, Cd, Cr, Zn and Cu) concentrations and stable Pb isotope ratios in rice plants exposed and unexposed to highway traffic pollution in Eastern China and found that Pb in rice plants were partly from the foliar uptake and considerable amount of foliar absorbed Pb and Cd were transported to rice grain.
Point 3. The article is well written, easy to read and logically organized
Response 3: Thank you very much for your extremely positive comment. We have considered all your suggestive comments and have highlighted the modified contents in the revised manuscript and response letter.
Point 4. The discussion around the contribution induced by the 3-stage quantitative model could be deepened, for example by a comparison of the results obtained with the 2 types of models. This would have made it possible to highlight the precision obtained
Response 4: We agree with your valuable comments and have added a section “ 3.4.3. Summary about three-stage quantitative analysis results” to the discuss this question. The content added is as follows:
3.4.3. Summary about three-stage quantitative analysis results
In this study, we provided a new attempt to determine the relative contributions of primary sources in grains based on Pb isotope analysis and model calculation using a three-stage quantitative analysis program involving (1) direct-source apportionment in grains with a two-end-member model, which can determine the contribution rate of soil and dustfall Pb contamination to grain. (2) source apportionment for soil and dustfall using the IsoSource model. (3) integration of results of (1) and (2) through multiplying and adding to finally get the contribution rate of all primary-sources for grains.
In the first process of direct source apportionment of grains, two-end-member model was commonly used for pollution receptors with only two pollution sources. Pb in pollution receptors can be regarded as a mixture of two main sources and the relative contribution rate of pollution sources to Pb in pollution receptors can be calculated ac-cording to two-end-member model. In this study, direct-source apportionment in wheat or rice grains, which there were only two direct Pb sources, soil and dustfall, was calculated by the two-end-member. For wheat grains, the source contribution rates of dustfall and soil to grains Pb pollution were respectively ~91% and ~9% and ~78% and ~22% for rice grains. Atmospheric dustfall was found to be the main direct source of Pb contamination in grain crops.
In the second step of source apportionment for soil and dustfall, which respectively have six and three sources, IsoSource model was used for the calculation of source contribution rate. For soil, the mean contribution rate of atmospheric dustfall was highest (66.7%), with dustfall thus being the major contributor to soil Pb contamination. Geological sources and irrigation water each contributed a marginal ~8%. K (6.5%), N (6%), and O fertilizers (5.5%) together contributed 18%. For dustfall, the mean order of source contribution rates was ranked as coal dust (69%) > geological sources (23%) > vehicular emissions (8%), which thus make coal dust the major source and vehicular emissions a marginal source.
In the third process of primary-source apportionment for grains, sequential multiplication and addition were used for this calculation. For wheat grains, all types of primary source to wheat Pb contamination, ranked as coal-fired industries (67.2%) > geological sources (22.6%) > vehicular emissions (8%) > potassium fertilizer (0.6%) ≈ irrigation water (0.6%) > organic fertilizer (0.5%) ≈ nitrogen fertilizer (0.5%). For rice grains, all types of primary source to rice Pb contamination source contributions of coal-fired industries (64.1%) > geological sources (22.8%) > vehicular emissions (7.6%) > irrigation water (1.6%) > potassium fertilizer (1.4%) ≈ nitrogen fertilizer (1.4%) >organic fertilizer (1.2%). These results indicated that source of coal-fired industries account for the largest proportion, of up to 60%. The control of coal-fired industries is the most important means of reducing Pb contamination of grain crops.
Through three-stage quantitative model analysis, not only the contribution rates of direct sources for grain were obtained, but also the contribution rates of primary sources for grain were calculated clearly. The results of each step obtained in the three steps can make us understand the detailed process of grain Pb pollution, and also hereby guiding the further practices of prevention and control for grain Pb pollution.
Detailed comments:
Point 1: Introduction
The bibliography must be deepened. It will also support the discussion
Lines 40: there are a lot of publications on industrial sites too.
Response 1: Thank you very much for your valuable suggestion. We have deepened the introduction and some sentences have been added to the text in red. And also, we have changed the sentence in line 40 into “Most investigations to date have focused on assessing contamination level and sources of Pb in soils and atmospheric particles”, and relevant references were added in this sentence.
Point 2: Materials and methods
All the analysis protocols for non-soil samples are missing
How atmospheric dustfall have been collected?
Response: Thank you very much for your significant question. We have added the analysis method of grain and dustfall in the section of “2.1 field sampling” and “2.2 Sample testing”. The added contents have been marked in red.
For statistics, the technical aspect is described. On the other hand, the description and the justification of the statistical tests carried out are missing.
Response: Thank you very much for your significant question. We have added the detailed description on the statistics. The added contents have been marked in red.
Line 153: what is the reference?
Response: Thank you very much for your significant question. The reference was “Huang, M., et al., Heavy metals in wheat grain: Assessment of potential health risk for inhabitants in Kunshan, China. Science of the Total Environment, 2008. 405(1-3): p. 54-61.” We have added it in the text.
Point 3: Results and discussion
Lines 166 and following and lines 175 and following: are the same paragraphs with different references
Response: Thank you very much for your significant question. We have deleted the repeated sentences in lines 175 and following in the added version of text.
Line 206: How were the concentrations in the grains analysed?
Response: Thank you very much for your very significant question. We have added the analysis method of grain in the section of “2.2 Sample testing”.
Line 208: it would be useful to know Pb-s, Pb-Sa, Pb-d
It would be beneficial for you to indicate the results of the concentrations in the soils and to comment on them in terms of the degree of contamination.
Response: Thank you very much for your very valuable comments. We have added the results analysis on the total concentration of Pb in soil (Pb-s), the available concentration of Pb in soil (Pb-sa) and the concentration of Pb in dustfall in the section of “3.1 Pb levels and risk in soil and dustfall”. In addition, the degrees of contamination in the soils and dustfall were also estimated in this section.
3.1. Pb levels and risk in soil and dustfall
A wide range of Pb total concentrations was observed in soils across the study area (Table 1). The Pb concentrations exceeded background values in 94% of the 32 soil samples with mean Pb being higher than their respective background values. Comparison of the mean Pb concentrations with the Chinese standard for agricultural soil indicated that Pb did not exceed the standard value. The concentration of available Pb indicated their degree of mobility in soil. The mean ratio of available fraction to total concentration of Pb was 67.4%. Tao et al. used the same method to extract the available Pb in soil. The results showed that mean ratio of available fraction to total concentration of Pb was about 20% [40]. Yang et al. studied the bioavailability and ecological risk assessment of heavy metals in soils around a mining area in Xinjiang, China and found that the ratio of available fraction to total concentration of Pb was 9% [41]. These results indicated that the mobility of Pb and therefore the greatest potential bio-availability in this study was relatively high.
Mean concentrations of Pb in dustfall was 223.86 mg/kg, significantly higher than background values. In comparison, concentration of Pb in Shijiazhuang city was 140 mg/kg [42], lower than that in this study. Also, Pb concentration in dustfall was similar with those measured in Shanghai [43]. The mean Pb concentration in the dustfall was significantly higher than background values, with almost nine times higher than the background value. These results all indicated relatively high degrees of contamination in dustfall of this region.
According to equation (1) for ecological risk, the mean Pi values of Pb for soils and dustfall were 1.52 and 8.54, respectively. The result indicated that Pb contamination level was higher in dustfall than in soil, with both exceeding the warning value (0.7), suggesting an extremely high ecological risk and pointing to the need for careful control of Pb pollution.
Table 1. Pb concentrations and Pi in soil and dustfall in the study area (mg/kg).
|
Parameter |
Samples |
|
Minimum value |
Maximum value |
Mean |
SD |
|
Concentration |
Soil (n = 32) |
Pb-s |
21.26 |
141.72 |
39.78 |
20.61 |
|
Pb-sa |
12.51 |
91.84 |
26.84 |
13.61 |
||
|
Dustfall (n = 10) |
|
73.82 |
328.97 |
223.86 |
76.33 |
|
|
Background values of Jiangsu |
|
- |
- |
26.2 |
- |
|
|
Chinese standard for agriculture soil |
|
- |
- |
300 |
- |
|
|
Pi |
Soil (n = 32) |
|
0.81 |
5.41 |
1.52 |
0.79 |
|
|
Dustfall (n = 10) |
|
2.82 |
12.56 |
8.54 |
2.91 |
Note: Pb-s = concentration in soil; Pb-sa = available Pb in soil; Pb-d = concentration in dustfall; “-” represents no reference value.
Figure 1: How do the 2 components of the ratio vary with respect to each other?
Response: Thank you very much for your significant question. The Pb BCF of grains in Figure 1-B was closely related to the Pb concentration of grains in Figure1-A. BCF can described the translocation capacity of Pb from the soil to the grain of crop plants. It was calculated as follows: BCF= Cgrains / Csoil. It can be seen from the calculation formula that BCF value was positively correlated with the concentration of Pb in grain and negatively correlated with the concentration of Pb in soil. Pb BCF values in the two grain types were similar, indicating that the translocation capacities of Pb from the soil to the two types of grain were similar. The reason for difference of Pb concentration (Figure 1-A) and the same BCF values (Figure 1-B) in two grains was that the Pb concentrations of the wheat soil and rice soil were different.
Line 225: How the concentrations of Pb in dust were analysed
Response: Thank you very much for your significant question. We have added the analysis method of dustfall in the section of “3.1. Pb levels and risk in soil and dustfall”. The added contents have been marked in red.
Line 253: Indeed, lead is a cumulative toxicant
Response: Thank you very much for your valuable comment. It was a question which deserved serious consideration. Indeed, Pb is a cumulative toxicant. Adults accumulate more Pb than children under the same exposure conditions. Logically, the health risk of adult caused by Pb pollution is higher than that of child. However, health risk assessment model considers more about vulnerability. Children may have weak resistance to Pb pollution compared with adults, and very low concentration will cause high health risk. This is really a very good question, which deserves serious consideration in the following studies.
Line 257: For the calculation it looks like you have chosen the average concentration (0.5 and 0.4 mg / kg, respectively). It seems that for the maximum concentration, the HQ for rice could be close or even greater than 1
Response: Thank you very much for your extremely valuable comment. Indeed, we have chosen the average concentration (0.5 and 0.4 mg / kg, respectively) for the calculation of HQ. the HQ for crop could be close or even greater than 1, when using maximum concentration. We have added the results analysis in the section of “3.3 Pb risk assessment of grain” in red text. The added section is presented below:
However, the average concentrations (0.54 for wheat grain and 0.45 mg/kg for rice grain, respectively) were used for the calculation of HQ. If the maximum concentration of grains was used for calculation, the HQs of grains, especially for rice grains, could be close or even greater than 1. Thus, remediation should still be undertaken to prevent further risks.
Lines 298: which vegetables? This influences a lot, depending on the very different accumulation capacities
Response: Thank you very much for your valuable comment. In line 298, the vegetables included radish, lettuce, ginger and chives. As you say, the factors of Pb isotopic compositions of above-ground (stem and leaf) and below-ground (root) parts of vegetables (and soils) influences a lot, depending on the very different accumulation capacities. Therefore, the species of vegetables need to be explained. Thus, we have added the types of vegetables in this sentence of the paper.
Shang studied Pb isotopic compositions of above-ground (stem and leaf) and be-low-ground (root) parts of vegetables including radish, lettuce, ginger and chives in the Chengdu area.
Line 343: it is necessary to describe what was done
Response 3: Thank you very much for your valuable question. We have added the description of parameter setting in the process of model operation to the section of “2.3.3. Source apportionment model” of the text in the red. The added sentences were as follows:
Before running the model, two parameters need to be set: one is the source increment, which is generally set to 1%; the other is mass balance tolerance. If it is set to 0.1 ‰, it means that the difference between the sum of weighted isotopic values of each pollution source and the isotopic values of the acceptor does not exceed 0.1 ‰, then the proportional combination is considered as optimal solution (Gregg 2003).

Round 2
Reviewer 3 Report
The article has been optimized.Some points still deserved more explanation and justification.
However, this document can be published as it is.